# Comparison between Effects of Retroactivity and Resource Competition upon Change in Downstream Reporter Genes of Synthetic Genetic Circuits

**DOI:** 10.3390/life9010030

**Published:** 2019-03-26

**Authors:** Takefumi Moriya, Tomohiro Yamaoka, Yuki Wakayama, Shotaro Ayukawa, Zicong Zhang, Masayuki Yamamura, Shinji Wakao, Daisuke Kiga

**Affiliations:** 1Department of Computational Intelligence and Systems Science, Tokyo Institute of Technology, Yokohama, Kanagawa 226-8503, Japan; tkfmmry@gmail.com (T.M.); zzhang@ali.c.titech.ac.jp (Z.Z.); my@c.titech.ac.jp (M.Y.); 2Department of Electrical Engineering and Bioscience, Waseda University, Shinjuku, Tokyo 169-8050, Japan; ru10-deka@suou.waseda.jp (T.Y.); mly-jp.klws@fuji.waseda.jp (Y.W.); wakao@waseda.jp (S.W.); 3Waseda Research Institute for Science and Engineering, Waseda University, Shinjuku, Tokyo 169-8050, Japan; shotaroayukawasb@gmail.com

**Keywords:** synthetic biology, systems biology, retroactivity, resource competition, reporter gene, protein-binding site

## Abstract

Reporter genes have contributed to advancements in molecular biology. Binding of an upstream regulatory protein to a downstream reporter promoter allows quantification of the activity of the upstream protein produced from the corresponding gene. In studies of synthetic biology, analyses of reporter gene activities ensure control of the cell with synthetic genetic circuits, as achieved using a combination of in silico and in vivo experiments. However, unexpected effects of downstream reporter genes on upstream regulatory genes may interfere with in vivo observations. This phenomenon is termed as retroactivity. Using in silico and in vivo experiments, we found that a different copy number of regulatory protein-binding sites in a downstream gene altered the upstream dynamics, suggesting retroactivity of reporters in this synthetic genetic oscillator. Furthermore, by separating the two sources of retroactivity (titration of the component and competition for degradation), we showed that, in the dual-feedback oscillator, the level of the fluorescent protein reporter competing for degradation with the circuits’ components is important for the stability of the oscillations. Altogether, our results indicate that the selection of reporter promoters using a combination of in silico and in vivo experiments is essential for the advanced design of genetic circuits.

## 1. Introduction

Reporter genes are important for the measurement of gene expression and have played a vital role in the advancement of molecular biology [1,2]. Downstream reporter genes are used to quantify the activity of upstream regulatory genes. A reporter gene comprises a promoter sequence, whose expression can be regulated by binding with a regulatory protein, and a coding sequence, whose product enables the quantification of gene regulation. Proteins encoded by reporter genes should be easy to detect and quantify. For these reasons, researchers use common reporter genes encoding β-galactosidase (β-gal) [3], chloramphenicol acetyltransferase (CAT) [4] and green fluorescent protein (GFP) [5] because the encoded proteins can be easily detected and quantified using enzymatic, drug resistance/enzymatic and fluorescence assays, respectively.

Reporter proteins also allow researchers to design, build, test and redesign synthetic genetic circuits, including the upstream regulatory genes. In synthetic biology, the dynamics of living cells with the mathematically-designed built-in circuits have been confirmed by testing the quantification of reporter proteins [6]. Based on the test results, synthetic biologists have redesigned synthetic genetic circuits to regulate the dynamics of living cells [7,8,9,10,11].

With a name of retroactivity, however, the dynamics of the upstream genes have shown to be disrupted in unexpected ways by a connection of downstream reporter genes with regulatory-protein binding sites that compete with the same binding sites of upstream regulatory circuits for accommodation of regulatory proteins [12,13,14,15,16,17,18,19,20,21,22,23]. For example, our modeling has shown that oscillation of genetic circuits (Figure 1A,E) can be disrupted by a connection of a reporter gene through retroactivity (Figure 1B,F). The genetic circuits shown in Figure 1A–D share upstream genes for which topological characteristics are favorable for oscillations. These circuits, however, differ in their reporter genes, which also have binding sites for regulatory proteins (Figure 1E–H).

In protein production processes, binding sites in promoters of both upstream regulatory and downstream genes compete for regulatory proteins (Figure 1F–H). The presence of additional downstream reporter genes increases the copy numbers of binding sites for regulatory proteins in a cell. The increased copy number of protein-binding sites thus lowers the number of free molecules of the regulatory protein in a cell. The lower number of free regulatory protein decreases the binding probability to each of its binding sites in designed regulatory genes. Consequently, the lower binding probability of the regulatory protein to each of their binding sites in upstream promoters disrupts the upstream network comprising regulatory proteins. Furthermore, difference in the number of additional binding sites among circuits will cause difference in retroactivity.

Similar to the retroactivity from downstream protein-binding sites to upstream regulatory proteins during protein production, resource competition from a downstream reporter protein to a protease which degrades both the regulatory protein and reporter protein also affects the dynamics of upstream regulatory genes (Figure 2) [12,24,25,26,27,28]. We focused on the sharing of tag-specific protease in resource competition other than the other sharing of components because much fewer copy numbers of tag-specific protease existed in a cell than that of globally shared components, such as RNA polymerase and ribosome. Upon introducing an artificial circuit having tagged reporter and regulatory proteins, protease molecules with a limited copy number originally for a small amount of tagged endogenous proteins have to degrade mixtures of a large number of tagged-proteins (Figure 2). Thus, in a condition of the same limited number of tag-specific proteases, increases in tagged reporter proteins can affect the dynamics of upstream regulatory genes. Conversely, RNA polymerase and ribosome for gene expression has a large copy number to accommodate components for artificial circuits with little perturbation in most cases.

In the present study, through a synthetic biology approach with a combination of in silico and in vivo experiments which cover a multi-dimensional parameter space, we determined how differences in regulatory protein-binding sites alter the dynamics of genetic circuits via changes in promoter activity and copy numbers of proteins degraded by a limited amount of proteases. We modeled and simulated alternative downstream genes in the Smolen oscillator, which forms positive and negative feedback loops (Figure 1A) [11,29,30]. These computations indicate that competition from protein-binding sites to their regulatory proteins and from target proteins to their tag-specific proteases can alter the dynamics of cellular systems. These two processes are, respectively, consistent with the previously proposed concepts of retroactivity [13,14,15] and competition between target proteins and their tag-specific proteases [24,26].

We then confirmed our prediction with microscopy using two cell strains that have the same regulatory circuit but different structure of regulatory protein-binding sites in the reporter gene (Figure 1B,C). We additionally modeled the dynamics of another circuit with the same number of protein-binding sites as the circuit shown in Figure 1B but with different allocation of those sites (Figure 1D). Such same number in the binding sites will not cause retroactivity. Even with the same number of protein-binding sites (Figure 1B,D), however, we found that different perturbations in upstream regulatory circuits are caused by different competitions from target proteins to their tag-specific proteases.

In further designs of genetic circuits, we included resource competition and compared it in electric circuits using established design strategies. Our results implied that comparison between the effects of retroactivity and resource competition such as protease sharing induced by the addition of reporter genes is important in the design of genetic circuits.

## 2. Materials and Methods

### 2.1. Mathematical Modeling of Chemical Reactions, Simulation and Stability Analysis

To predict the dynamic behaviors of the components in the genetic circuit models, we constructed mathematical models based on biochemical reactions (e.g., the dynamics of interactions with the promoter, protein synthesis and decay of the components). Our mathematical models were mostly based on previous papers [11,12,29,30]. We modified the previous model based on recent findings and measurements where one LacI tetramer protein formed a DNA loop, not two LacI tetramer proteins (Appendix A) [31,32,33]. We thus defined the loop dissociation rate (*k_ul_*) [34], the loop forming rate (*k_l_*) and the loop unforming rate (*k_-l_*) [31]. Binding rates (*ka*) at the AraC protein binding site were determined in a previous study, and these indicated that IPTG weakly inhibits arabinose-binding to AraC [11,35]. Detailed mathematical modeling and simulation methods are described in the Appendix A. All parameters are described in Appendix B.

We determined the stability of equilibrium points with linear stability analysis; we formulated the nonlinear ODEs (Ordinary Differential Equations) based on Equations (1)–(76) described in the Appendix A. The equilibrium points were calculated by solving these simultaneous ODEs using the Newton–Raphson method. If all the eigenvalues of Jacobian linearization around the equilibrium points have negative real parts, the equilibrium points are stable. To confirm whether all the eigenvalues of the Jacobian matrix have negative real parts, we used modified Routh’s method [36].

Finally, we plotted the stable fix points and oscillation amplitude using the process described below, in parameter spaces for all inducer conditions (arabinose: 21 points from 0.01% to 1.0% at logarithmically spaced values; IPTG (isopropyl β-d-1-thiogalactopyranoside): 46 points from 0.001 to 31.6 mM at logarithmically spaced values). We plotted stable fix points determined by the linear stability analysis described above. In unstable areas, particularly, we plotted each of the amplitude values from ODEs simulations using the pseudocolor “jet” of MATLABs built-in color.

### 2.2. Bacterial Strains and Construction of Plasmids

*Escherichia coli* strains and plasmids used in this study are listed in Appendix C. Transductant JS006 (MG1655 Δ*araC* Δ*lacI* Kan^S^) was kindly provided by the Hasty lab [11]. As described in Appendix C, pJS167 plasmid was modified to create pJSDT267 with an alternative reporter, Plac-*gfp*. To quantify the maximum transcription activities of the reporter *gfp* gene driven by the lac/ara and lac promoters, pJSDT171 and pJSDT271 plasmids (Appendix A) were constructed as described in Appendix C, using pJS167 and pJSDT267, respectively.

### 2.3. Reporter Assay

Reporter assays were performed as described previously [11], with slight modifications. Overnight cultures of reporter strains grown at 37 °C in LB medium containing the appropriate antibiotics were diluted to an OD_600_ of 0.1 in medium. The diluted cultures with or without inducers (i.e., 0.1% arabinose) were incubated at 37 °C for 2 h. After incubation, 1.0 mL of each culture was washed with phosphate-buffered saline by centrifugation, and the raw fluorescence intensity was measured with a flow cytometer (FACSCalibur; Becton-Dickinson, Franklin Lakes, NJ, USA) with excitation at 488 nm and emission at 515–545 nm. A strain containing only the Ptet-*gfp* plasmid, which constitutively expresses GFP, was used as a positive control in our reporter assay and a strain containing the PBAD/ara plasmid, which does not express GFP, was used as a negative control.

### 2.4. Microscopy Experiments

All images were acquired using an Eclipse Ti-E inverted microscope (Nikon Instruments Inc., Tokyo, Japan) with an Apochromat lens (CFI Plan Apo 40x objective lens; Nikon Instruments Inc., Tokyo, Japan) and EMCCD camera (iXon3 897; Andor Technology Ltd., Belfast.). Images were analyzed using NIS-Elements Advanced Research software (Nikon Instruments Inc., Tokyo, Japan). Differential interference contrast (DIC) images were obtained using a halogen lamp (HLL 12V 100W; Nikon Instruments Inc., Tokyo, Japan). Fluorescence images were obtained using a mercury vapour lamp (Intensilight C-HGFIE; Nikon Instruments Inc., Tokyo, Japan) and a GFP filter cube (GFP-HQ filter cube, EX455-485, DM495, BA500-545; Nikon Instruments Inc., Tokyo, Japan).

To prepare samples for microscopy, overnight cell cultures (JS00611 or JSDT10611) were diluted 100-fold into 3.0 mL of fresh LB medium with antibiotics (50 μg/mL ampicillin and 30 μg/mL kanamycin). Cells were grown at 37 °C for 30–60 min for the cell population to reach a sufficient density (OD_600_ = 0.2). After the addition of the specified concentrations of each inducer into 1 mL of cell culture, 400 μL of the culture was placed between the top of a glass plate and the bottom of a 2% agarose pad (about 12 mm in diameter and 3 mm thick), which also contained the medium with specified inducers. After 30 min of incubation at 37 °C in a stage-top incubator for microscope (Thermo Plate; Tokai Hit Co., Ltd, Shizuoka, Japan), images were collected in the DIC and GFP fluorescence channels every 3 min for 3 h.

### 2.5. Imaging Process

For the analysis of oscillation dynamics, we performed microcolony recognition, background subtraction and fluorescence quantification (Appendix A) using MATLAB software (MathWorks, Natick, MA, USA), ImageJ [37,38] and the ImageJ plugin MTrackJ [39]. From the above process, we collected fluorescent time-courses determined by the backtracking of ROI (Region of Interest) adjusted to the microcolony-growth. These fluorescent time-courses allowed us to plot heat maps of oscillation damping on each concentration of the two inducers. Detailed methods are described in the Appendix A.

## 3. Results and Discussion

### 3.1. Mathematical Modeling and Simulation Suggested Perturbation of Oscillation Dynamics by Competitions for Regulatory Protein and Protease

By quantifying reporter gene expression, researchers have analyzed the dynamics of the Smolen oscillator, which forms positive and negative feedback loops (Figure 1A,E) [11,29,30]. Both upstream regulatory genes (*araC* and *lacI*) are driven by the lac/ara promoter, which contains protein-binding sites for one AraC activator and two LacI repressor protein molecules (Appendix A) [40].

Similar to our previous modeling study where the addition of a reporter gene to upstream genes changed oscillation dynamics of those circuits [12], we compared a genetic circuit with a modified one containing the replacement of a protein-binding site in a downstream reporter promoter region (Appendix A). Although upstream regulatory genes in living cells used in the previous oscillation study were accompanied by the downstream reporter *gfp* gene driven by the lac/ara promoter (lac/ara-reporter circuit; Figure 1B,F), the original modeling of the oscillator excluded the binding of regulatory proteins to their protein-binding sites in the reporter gene [11]. In contrast, we recently demonstrated the effects of retroactivity and protease sharing by the addition of the reporter protein using mathematical model [12]. In this study, we also modeled the dynamics of the Smolen oscillator connected to another reporter *gfp* gene driven by the lac promoter (lac-reporter circuit; Figure 1C,G, Appendix A). The detailed mathematical modeling is described in Appendix A, the Methods Section 2.1, Appendix B and the Appendix A.

Upon the removal of AraC-binding sites in downstream reporter genes, simulation of the lac-reporter circuit resulted in a narrowed oscillation area for each concentration of the two inducers (arabinose and IPTG), compared with that of the lac/ara-reporter circuit (Figure 3), even though the lac/ara and lac promoters have the same maximum transcription rate in our modeling. We plotted time-courses of the lac/ara-reporter and lac-reporter circuits by ODEs simulations under high arabinose conditions (Figure 3A–D). For some sets of parameters, the variables for numbers of molecules per cell fell below 1. Thus, the stochastic dynamics of molecules possibly regulate oscillations in our living cells. In Smolen’s oscillators, however, stochastic Gillespie and deterministic simulations showed similar periods for a set of parameters [11,41]. From the stability analysis of parameter spaces for the two inducers, the lac-reporter circuit showed a narrower oscillation area than the lac/ara-reporter circuit (Figure 3E–F). In particular, upstream oscillation dynamics in the lac-reporter circuit showed a shorter period with high arabinose and low IPTG concentrations than those in the lac/ara-reporter circuit (Figure 3B,D). We also found, with high arabinose and IPTG concentrations, that the lac-reporter circuit remained fixed, although the lac/ara-reporter circuit oscillated (Figure 3A,C,E,F). Thus, due to differences in regulatory protein-binding sites in downstream reporter genes only, the lac/ara- and lac-reporter circuits showed differences in their dynamics, although the two circuits share the same maximum transcription rate for the reporter gene and the same structure for the regulatory circuit. This is an example of in silico retroactivity [13,14,15] and/or resource competition [24,26], in other words, titration of the component and/or competition for degradation, respectively.

### 3.2. Microscopy Experiments Showed that Downstream Reporter Genes Changed the Oscillation Dynamics of Upstream Regulatory Genes

Similar to our simulations in Figure 3, our microscopy experiments with the same set of inducer concentrations showed oscillation of the lac/ara-reporter *E. coli* strain (Figure 4A and Appendix A) and no oscillation of the lac-reporter *E. coli* strain (Figure 4C and Appendix A). In contrast, both strains oscillated under another set of inducer concentrations (Figure 4B,D and Appendix A). In other words, our microscopic analysis showed that differences in the copy number of protein-binding sites in downstream components alters the dynamics of upstream components, as previously reported [19,20].

Before performing a detailed analysis of the microscopic observations, we compared the maximum transcription activities of the lac/ara and lac promoters (Figure 5 and Appendix A). Our reporter assays using living cells revealed that the maximum transcription activity of the designed lac promoter results only in a 1.3-fold higher accumulation of GFP than that of a previously developed lac/ara promoter (Figure 5A) [40]. Notably, the following in silico analysis showed that this 1.3-fold difference was ignored in this study. From the in silico analysis, when the maximum transcription activity of the lac/ara promoter (*αb_d_*_1_) and lac promoter (*b_d_*_2_) is the same, we can see a clear difference in the oscillation area (Figure 5B,C). Furthermore, this difference remained after a ≥1.3-fold change in the maximum transcription activity of the downstream lac promoter (Figure 5C–E). Thus, the 1.3-fold higher accumulation of GFP from the lac promoter, compared with that from the lac/ara promoter, could not be the main source of the difference in the in vivo oscillation dynamics affected by retroactivity and/or protease sharing from each of the two promoters.

Our detailed microscopic analysis of the lac/ara- and lac-reporter circuits showed different sensitivities in the oscillation against changes for parameter spaces in the inducer concentrations (Figure 4), as predicted by our simulations in the previous section (Figure 3). Considering that even the strain with the constitutive expression of GFP showed weak fluctuations in GFP levels (Appendix A) [42], we needed a collective evaluation of data from each experimental condition. We therefore counted several oscillation bottom points of a time course, as outlined in Appendix A, the Methods Section 2.5 and the Appendix A. From thousands of time courses (Appendix A), our evaluation showed an apparent difference in the oscillation tendency between constitutive GFP expression (Ptet-*gfp* strain) and the strongly oscillating expression reported in the previous work (lac/ara-reporter circuit strain) [11]. Thus, the accumulated microscopy time-lapse data were used for the oscillation evaluation based on the bottom count (Figure 3, Appendix A).

Intriguingly, similar effects of inducer conditions on oscillation damping were found in both the comprehensive simulation (Figure 3; Figure 5), and imaging process with comprehensive microscopic experiments (Figure 4, Appendix A) (refer to the Methods Section 2.5 and the Appendix A). First, both strains showed stabilized oscillation with high arabinose and low IPTG (Figure 3B,D–F, Figure 4B,D,F,H–J, Appendix A). With low arabinose, neither strain showed clear oscillation (Figure 3E,F, Figure 4I,J, Appendix A). On the contrary, with high arabinose and IPTG concentrations, the lac/ara-reporter strain shows clear oscillation (Figure 3A,E, Figure 4A,E,I, Appendix A), whereas the lac-reporter strain does not (Figure 3C,F, Figure 4C,G,J, Appendix A).

In the same microcolony on an agar plate, the physiological conditions of cells, even in a rim region of the microcolony, varied with time probably because of cell–cell communications or nutrient deficiency. Accordingly, in our experiments with long incubation periods, we could not detect oscillations in most microcolonies, even when these microcolonies included cells with oscillations at earlier time points (Appendix A). Thus, our microscopy experiments were limited to 3 h. Due to this limitation, we were not able make other measurements, such as those required to estimate coefficients of variation in the period.

### 3.3. Comparison of Retroactivity and Protease Sharing with Upstream Gene Expression Dynamics Even for the Same Set of Protein-Binding Sites in a Downstream Component Having Differences in the Allocation

The correspondence of parameter dependency for oscillation, found in the results from both our simulations (Figure 3 and Figure 6A,B) and microscopic observations (Figure 4), encouraged us to perform additional modeling to compare effects of retroactivity and protease sharing. One reason for the different oscillation dynamics between the lac/ara- and lac-reporter circuits could be the different copy numbers of total AraC-binding sites in the whole circuit (Figure 1B,C, Figure 3 and Figure 6A,B). Thus, we additionally modeled the modified lac-reporter plasmid with an AraC decoy site (lac-reporter + AraC decoy circuit) (Figure 1D and Figure 6C). However, although the modified circuit shares the same copy number of binding sites for the regulatory proteins with the lac/ara-reporter circuit (Figure 1B,D), the two simulated circuits show a difference in the oscillation dynamics in terms of regulatory protein amplitude (Figure 6A,C) and period (Appendix A). In other words, the difference between the dynamics of the lac/ara- and lac-reporter circuits (Figure 6A,B and Appendix A) remained even after the addition of the decoy site in the lac-reporter circuit, although this addition resolved the difference in the number of binding sites between the two circuits. Thus, we suspect that resource competition, not retroactivity, is a potential main source of the difference in the dynamics.

We then tried to examine two potential sources of different activities of the lac/ara-reporter circuit and the lac-reporter + AraC decoy circuit which shared the same number of protein-binding sites, featuring one of the two types of molecular competition: retroactivity and protease sharing (Figure 6 and Appendix A). In other words, we added simulations without competition from protein-binding sites to their regulatory proteins and/or that from target proteins to their tag-specific protease. Without both competitions, all three circuits showed the same oscillation area and regulatory-protein amplitudes (Figure 6J–L and Appendix A), although the lac/ara-reporter circuit showed a lower GFP amplitude in expression level than the others (Appendix A). Protease sharing, rather than retroactivity, mainly accounts for a combined effect of the protease sharing and retroactivity (Figure 6A–F); note that the combined effect in both of in silico and in vivo showed a good agreement (Figure 4I–J and Figure 6A–B). By addition of the protease sharing (Figure 6D–F), we found nearly the same effect with the combined effect (Figure 6A–F). Furthermore, the perturbation due to the protease sharing is dependent on GFP production rate of a circuit. The lac-reporter circuit and the lac-reporter + AraC decoy circuit, which have larger GFP production rate than the ara/lac-reporter circuit (Appendix A), showed larger perturbation (Figure 6E,F,K,L) than the lac-reporter circuit (Figure 6D,J). The higher GFP production in the lac-reporter circuit in simulation was also found in microscopy experiment (Figure 4). In addition to the effect in the oscillation amplitude, a similar essential effect of the protease sharing, rather than retroactivity, was also found in the oscillation period (Appendix A).

The large effect produced by protease-sharing can be explained in detail when we focus on a set of IPTG-arabinose concentrations. Although all circuits showed the same oscillation period, the much higher temporal GFP expression level profile of the lac promoter than that of the lac/ara promoter accounts for the large effect produced by protease-sharing (Figure 7A). This difference in expression level was derived from different temporal expression rate profiles of the two promoters (Figure 7B) whose expression rates depend on the concentration of regulatory proteins (Figure 7C,D). Despite the two promoters in the simulation having the same maximum transcription rate parameter (*K_a_* and *k_r_*), AraC concentration during the oscillation period is not enough to achieve the maximum expression rate from the lac/ara promoter.

On the other hand, the addition of retroactivity only slightly changed the oscillation area and amplitude (Figure 6G–L). In a detailed view of such simulation, a removal of AraC-binding sites from the lac/ara-reporter circuit showed small change in the fix area and increase in amplitude (Figure 6G,H), as shown in previous theoretical study [22,43]. (Figure 6G,H). This slight change was cancelled by the recovery of the AraC-binding sites as the decoy site (Figure 6G–I), not as a part of the promoter. This cancellation in simulation with retroactivity and without protease sharing can be explained by the promoter transition diagram of regulatory protein binding sites (Appendix A). In more simple words, the same numbers of binding sites of the lac/ara-reporter circuit and the lac-reporter + AraC decoy circuit makes the same fix area and amplitude.

### 3.4. Relationship of the Effects of the Molecular Competitions between Electric Circuit and Genetic Circuit

Molecular competition in our combined genetic circuit comprising the upstream and downstream genetic circuits corresponds to a competition for voltage between regulatory and downstream components in a combined electronic circuit (Figure 8A). These components share a DC power supply of the circuit. Using interconnected regulatory and downstream components, previous studies have found a similar situation between retroactivity on the dynamics of genetic circuits by addition of protein-binding sites on DNA and perturbation on the dynamics of an electronic circuit by connection of components [12,14,44]. Considering this similarity, we introduced a notion of “impedance” using a signal-amplifier circuit comprising regulatory and downstream components with dynamic voltage ranges (Figure 8A black part) [12]. A similar notion of admittance in genetic circuits was reported by Gyorgy and Del Vecchio using Thévenin’s theorem [14]. Here, we propose that the output-impedance of the shared DC power has isomorphism to the amount of a shared cellular component, such as the tag-specific protease.

A host cell with limited amount of a protease whose activity is shared among regulatory and downstream components corresponds to a DC power supply with high output impedance, which causes fragility upon the connection of additional elements (Figure 8B). For a regulatory circuit whose function is partially defined by the efficient degradation of regulatory proteins, the addition of a downstream reporter gene severely perturbs the circuit because the limited amount of protease is unable to efficiently degrade the additional reporter protein or the regulatory proteins, both of which have a specific tag for the protease. Such perturbation in degradation of regulatory proteins leads change in behavior of the regulatory circuit. This situation reminds us of a shared DC power supply with a high output impedance. In a system with a shared DC power supply and a regulatory component, current from the supply is divided into a main flow to the regulatory component and a subtle flow to the small internal resistance. Upon connection of downstream components, a new current for the downstream components dramatically changes the current to the regulatory component.

Increased protease concentration which requires additional energy for protease production circumvents the competition from target proteins to the proteases (Figure 8C). Simultaneously, the increased degradation rates of the regulatory proteins require extra cost for increased production rate if equivalent amounts of regulatory proteins are required. Such trade-off between robustness and energy has similarity with an electric circuits driven by the shared DC power supply with low output impedance. In this case, the current from the power supply is divided into the regulatory component, downstream component, and internal resistance, which consumes most of the power. Due to such large consumption other than regulatory and downstream components, difference in the downstream component hardly affects the activity of the regulatory circuit. In other words, both biological and electrical circuits with such a shared supply can be robust with extra energy consumption. In very recent studies, oscillator dynamics were shown to depend on the fine-tuning of product degradation pathways [45,46]. Controlled timing of degradation may require more energy than maintenance of constant expression levels of degradation enzymes. Questions arising from this hypothesis will be important subjects of future studies. In other aspects, biological insulators proposed for robust design of circuits [13] also seem to require extra energy. In addition to a trade-off in impedance-matching upon connection of components consisting of combined electric circuits, the trade-off upon constriction of genetic parts consisting of combined genetic circuits in living cells is important in the large-scale design of genetic circuits for synthetic biology. In living cells, other shared sources such as metabolic systems can be a shared power supply in electric circuits.

## 4. Conclusions

Between in silico and in vivo experiments, we here confirmed correspondence in parameter dependencies in the oscillation of either of the two genetic circuits that differ only in the regulatory-protein binding sites at the promoter of the reporter gene. Further detailed simulation allowed comparison of effects of retroactivity and protease sharing derived from competitions for the limited amount of protease that degrades the regulatory and reporter proteins. Even with the same number of protein-binding sites in genetic circuits, intriguingly, differences in resource competition alter perturbations in upstream regulatory circuits. Our study thus suggests the importance of the evaluation of retroactivity and protease sharing upon any addition of downstream genes to natural regulatory networks. Adequate design based on such evaluation of retroactivity and protease sharing accelerates the implementation of the desired properties in biotechnology, as well as deepens the understanding of natural genetic systems in bioscience.

## Figures and Tables

**Figure 1 life-09-00030-f001:**
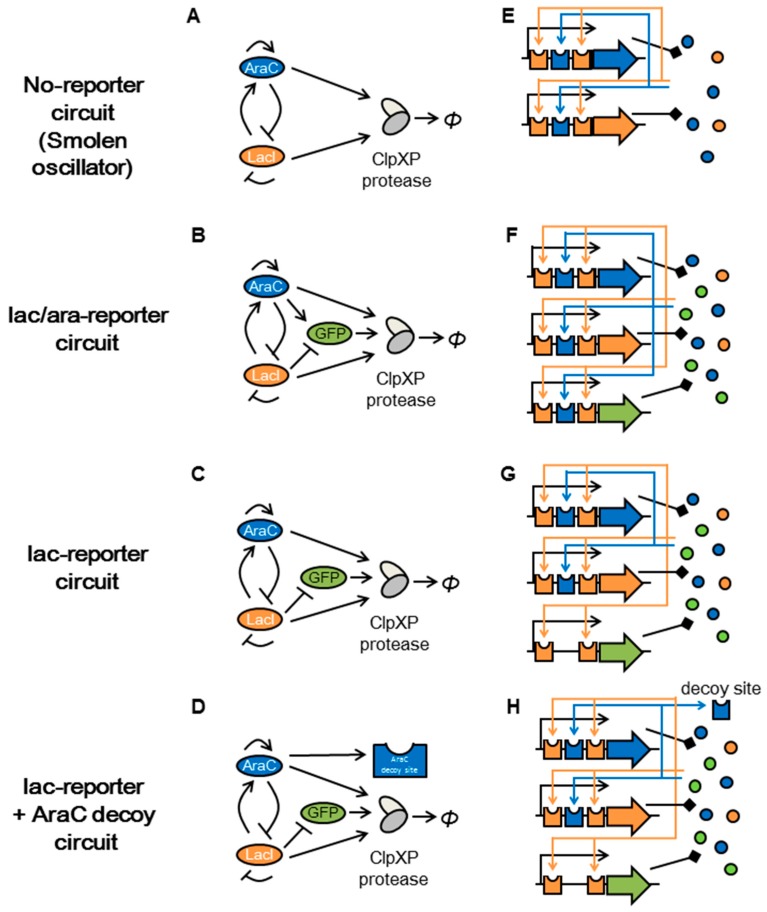
Schematic representation of retroactivity in the synthetic genetic circuits. AraC dimer protein molecules, LacI tetramer protein molecules and GFP monomer protein molecules are shown with blue, orange, and green circles, respectively. The genetic circuits (**A**–**D**) have the same structure of upstream regulatory genes in which each coding region (*araC* and *lacI*) was transcribed by the same lac/ara promoter: the AraC-activated and LacI-repressed promoter. (**A**) Schematic of the no-reporter circuit consisting of the Smolen oscillator. (**B**) Schematic of the lac/ara-reporter circuit consisting of the Smolen oscillator and a *gfp* reporter driven by the lac/ara promoter. (**C**) Schematic of the lac-reporter circuit consisting of the Smolen oscillator and a *gfp* reporter driven by the lac promoter. (**D**) Schematic of the lac-reporter + AraC decoy circuit consisting of the Smolen oscillator and a *gfp* reporter driven by the lac promoter with AraC decoy site. (**E**–**H**) Retroactivity from protein-binding sites to regulatory proteins. Regulatory genes in all models contain a promoter with one AraC protein-binding site and two LacI protein-binding sites. Binding of regulatory proteins to these sites is shown with stealth arrows (AraC blue; and LacI orange). Protein-binding sites on DNA are shown with dented rectangles. Protein-coding sequences on DNA are shown with arrows enclosed with black lines. Protein productions are shown with black-squared arrows. (**E**) No reporter gene. (**F**) The reporter gene contains a promoter with one AraC protein-binding site and two LacI protein-binding sites. (**G**) The reporter gene contains a promoter with two LacI protein-binding sites. (**H**) The reporter gene contains a promoter with two LacI protein-binding sites and an AraC decoy site with one AraC protein-binding site. The difference between the (**F**) and (**H**) models, which have reporter genes containing the same copy number of protein-binding sites, cause differential perturbations to upstream regulatory genes.

**Figure 2 life-09-00030-f002:**
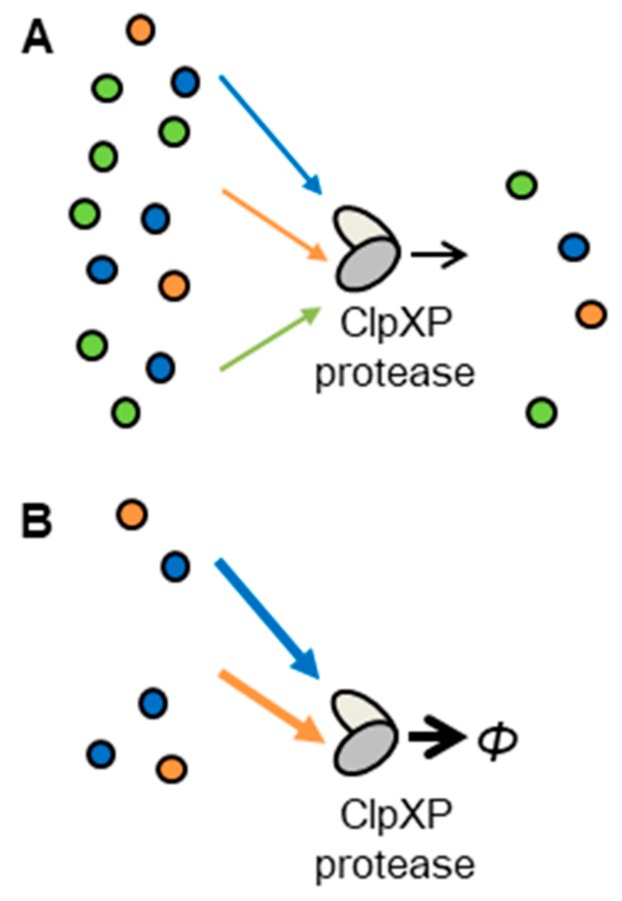
Schematic representation of protease sharing. AraC dimer protein molecules, LacI tetramer protein molecules and GFP monomer protein molecules are shown with blue, orange, and green circles, respectively. Protease sharing from tagged target proteins to tag-specific proteases. Tagged target proteins (AraC, LacI and GFP) are degraded by a ClpXP protease. Protein degradation by proteases is displayed with triangle-head arrows (AraC: blue, LacI: orange; and GFP: green). (**A**) By increases in reporter proteins or other proteins, degradation rates of upstream proteins are low. (**B**) Without reporter proteins or other proteins, degradation rates of upstream proteins are high.

**Figure 3 life-09-00030-f003:**
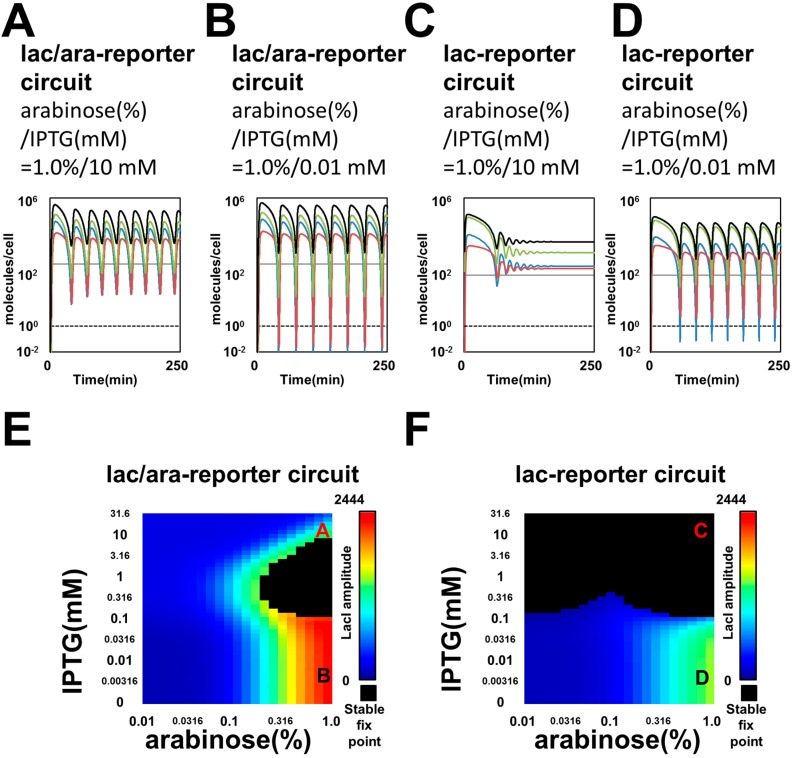
Deterministic simulation of the oscillatory synthetic genetic circuits containing reporter genes with alternative promoters. (**A**–**D**) Time courses show the numbers of free LacI tetramer proteins (orange), free AraC dimer proteins (blue) and free monomer GFP (green) and the sum of the three proteins, all of which have an SsrA tag for fast degradation (black). (**E**–**F**) The behaviors of the Smolen oscillators with respect to arabinose (*x*-axis) and IPTG concentrations (*y*-axis) are shown by deterministic simulation. Black regions show stable fixed points. The colors in the heat map demonstrate the amplitude of LacI in the oscillation.

**Figure 4 life-09-00030-f004:**
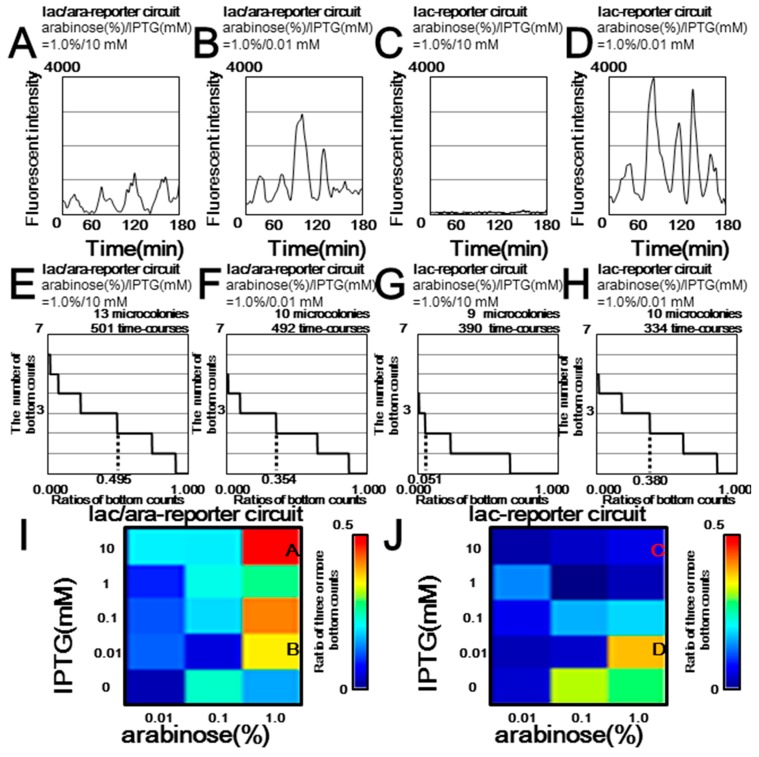
Microscopy observation of the oscillatory synthetic genetic circuits containing reporter genes with alternative promoters. (**A**–**D**) Representative time courses of the GFP fluorescence of cells. The fluorescence intensities are shown with black lines. The lac/ara-reporter circuit strain contains pJS167 and pJS169 plasmids. The lac-reporter circuit strain contains pJSDT267 and pJS169 plasmids. (**E**–**H**) Cumulative relative frequency distributions of the relative bottom counts. These distributions are shown concerning the ratios of bottom counts in descending order (*x*-axis) and the number of bottom counts (*y*-axis). The ratios of three or more bottom counts are shown below the distributions. Relative bottom counting was determined from the rate of three or more bottom counts, defined in Appendix A, the Methods Section 2.5 and the Appendix A. (**I**–**J**) The behaviors of the Smolen oscillators for the arabinose (*x*-axis) and IPTG concentrations (*y*-axis). The colors in the heat map show each relative bottom counting which corresponds to oscillation stability.

**Figure 5 life-09-00030-f005:**
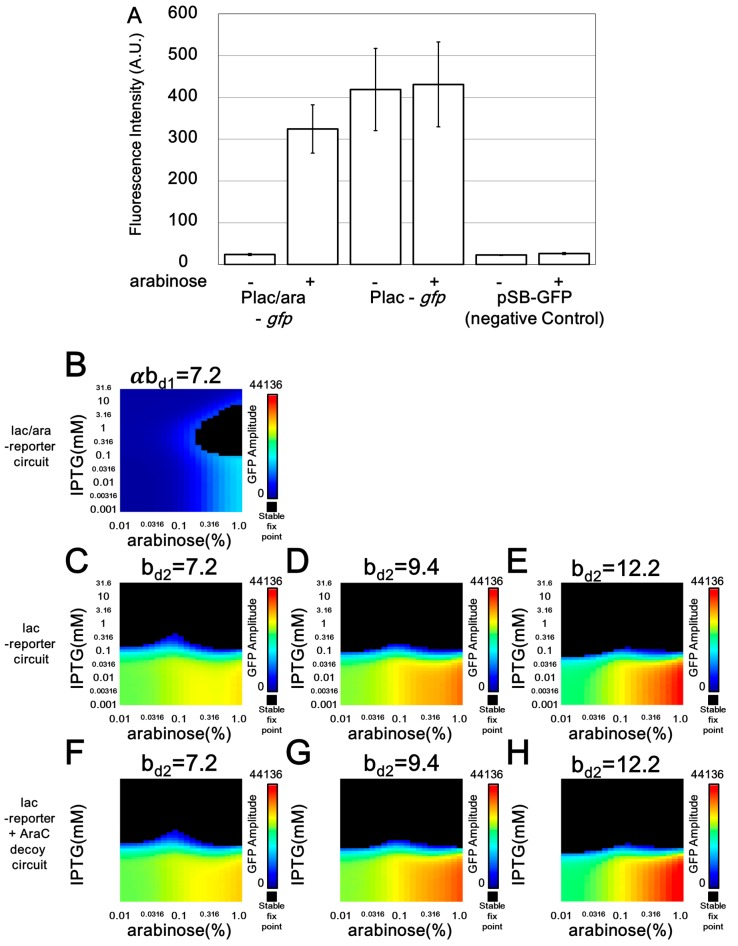
Effects by difference in maximum transcription rate of a reporter gene. (**A**) To quantitatively determine the performance of the lac/ara and lac promoters (pJSDT171 and pJSDT271 plasmids) in the presence or absence of arabinose, the fluorescence intensities of the reporter strains constitutively expressing AraC were measured. The assays were performed in quadruplicate. Error bars indicate the standard deviation. (**B**–**H**) The behaviors of Smolen oscillators with respect to arabinose concentration (*x*-axis) and IPTG concentration (*y*-axis) are shown by deterministic simulation. The colors in the heat map demonstrate the amplitude of GFP in the oscillation. Black regions show stable fixed points. αb_d1_ and b_d2_ represent maximum transcription rate of downstream lac/ara promoter and lac-reporter, respectively.

**Figure 6 life-09-00030-f006:**
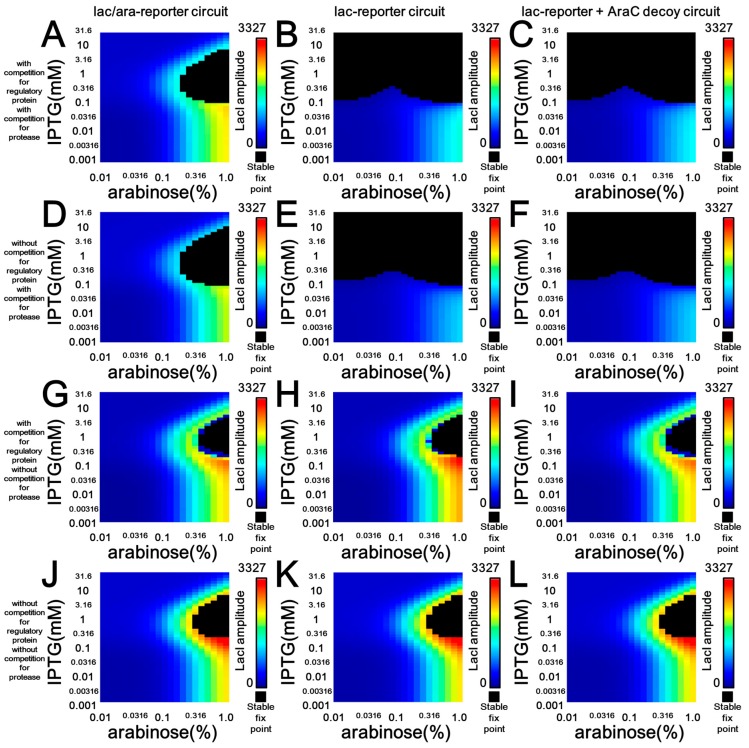
Oscillation stability in the presence or absence of downstream molecular competitions by deterministic simulation. Deterministic simulation shows the behaviors of the Smolen oscillators with respect to arabinose (*x*-axis) and IPTG concentrations (*y*-axis). The colors in the heat map demonstrate the amplitude of LacI in the oscillation. Black regions show stable fixed points. (**A**–**C**) Simulation with both downstream competitions from protein-binding sites to their regulatory proteins and from target proteins to their tag-specific proteases. (**D**–**E**) Simulation without downstream competition from protein-binding sites to their regulatory proteins and with downstream competition from target proteins to their tag-specific proteases. (**G**–**I**) Simulation with downstream competition from protein-binding sites to their regulatory proteins and without downstream competition from target proteins to their tag-specific proteases. (**J**–**L**) Simulation without downstream competition from protein-binding sites to their regulatory proteins and from target proteins to their tag-specific proteases.

**Figure 7 life-09-00030-f007:**
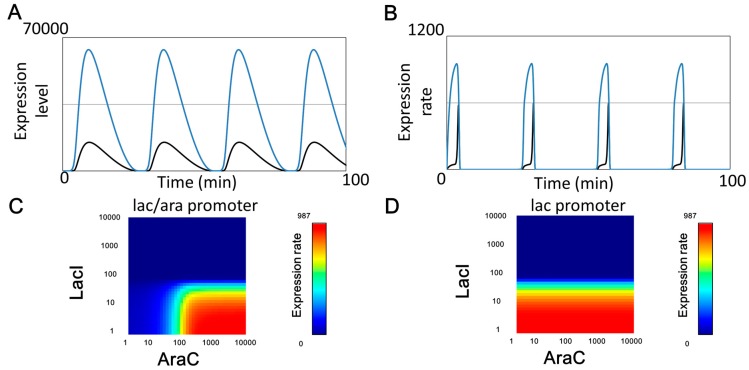
Oscillation timing of the three synthetic circuits. Deterministic simulations performed at arabinose 1.0% and IPTG 0.01 mM without downstream competitions from protein-binding sites to their regulatory proteins and from target proteins to their tag-specific proteases. The amount of active promoter of the lac/ara-reporter circuit model (black), the lac-reporter circuit model and the lac-reporter + AraC decoy circuit model (blue). (**A**) GFP Expression level of the three synthetic circuits. (**B**) Expression speed performed by the amount of GFP active promoter. (**C**) The lac/ara promoter state with black-lined trajectory at arabinose 1.0% and IPTG 0.01 mM. (**D**) The lac promoter state with black-lined trajectory at arabinose 1.0% and IPTG 0.01 mM.

**Figure 8 life-09-00030-f008:**
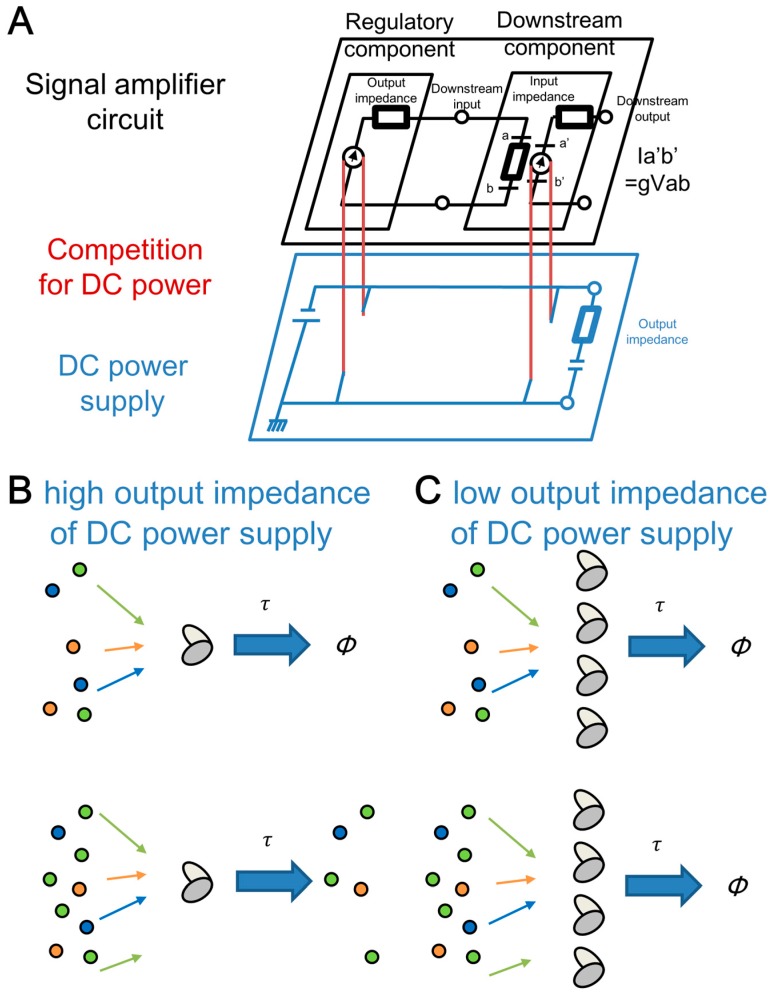
Relationship of the effects of the molecular competitions between electric circuit and genetic circuit. (**A**) The living cells with synthetic genetic circuits in our study behave as the electric circuits. In the upper layer, competition from protein-binding sites to regulatory proteins behave as impedance matching in signal amplifier circuits. In the lower layer, competition for the tag-specific protease behave as output impedance of DC power. (**B**) Few proteases correspond to a high output impedance of DC power supply. (**C**) Many proteases correspond to a low output impedance of DC power supply.

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
