# Peer review of "Comparison between Effects of Retroactivity and Resource Competition upon Change in Downstream Reporter Genes of Synthetic Genetic Circuits"

_life, 2019, doi:10.3390/life9010030_

Round 1

Reviewer 1 Report

See atatched file

Author Response

Note:

Blue-coloured text = the reviewers' comments

Black-coloured text = responses to the comments (starting with '=>') or unrevised text Underlined black-coloured text = text surrounding revisions

Green-coloured text = revised text

Underlined green-coloured text = revised text with correlation

 ------------------------------------------------------------------------------------

Responses to Reviewer.

The authors investigate two different effects that reporter promoter can have on synthetic gene circuits, which they term “retroactivity” and “resource competition”. In retroactivity, the reporter promoter binding sites compete with the regulatory binding sites (i.e. part of the circuit) for transcription factor. This has the effect of titrating the transcription factor (e.g. repressors, activators) and can change the activation/repression curve of the promoter (e.g. see Brewster et al. Cell 2016). This can increase the switching point between the promoter on and off state as well as sharpen that transition (i.e. increase the apparent cooperativity and make it more all or none). In resource competition, the reporter protein competes with the circuit in terms of cellular resources, such as transcription, translation and degradation of the components. In this manuscript, they consider only competition for degradation of the proteins by shared proteases.

The authors look at these two effects on the oscillations of the dual-feedback/Smolen oscillator (Stricker et al. Nature 2008) using a combination of experiments and simulations. Previously (Moriya et al. BMC Systems Biology 2014), the authors looked at the effect of having a reporter promoter (vs no reporter) in this circuit. In this study, they compare the three reporter setup shown below.

They show experimentally that removing binding sites for the activator AraC (going from reporter A to B) diminishes the fraction of the explored parameter space supporting oscillations. This is also confirmed in simulations. Finally, they conclude that this is due to competition for protease and not through titration of the activator, by simulating circuit C.

While potentially interesting, there are many severe issues preventing publication of this study.

=> We appreciate the reviewer for understanding our study and giving us a variety of useful comments and suggestions for this manuscript.

First and most importantly, the authors appear to contradict themselves. They first claim that reporter A and reporter B have similar maximum expression (~1.3 fold, Figure 6). Then, they verify in simulations that this difference cannot account for the different dynamical behavior between circuits with reporter A and B. Therefore, they conclude that difference in reporter expression causing protease competition (i.e. resource competition) could not explain the measured difference in dynamical behavior. (Line 255-258).

Thus, the 1.3-fold higher accumulation of GFP from the lac promoter, compared with that from the lac/ara promoter, could not be the main source of the difference in the in vivo oscillation dynamics affected by retroactivity and/or protease sharing from each of the two promoters.”

Then, towards the end, they conclude that:

Protease sharing, rather than retroactivity, mainly accounts for a combined effect of the protease sharing and retroactivity”

And that:

Furthermore, the perturbation due to the protease sharing is dependent on GFP production rate of a circuit.”

The fact that circuit A and C could have different oscillations dynamics but with the promoter having similar total expression would be very surprising. The only possibility is if the two reporters (A vs B/C) have a different temporal expression profile (but similar peak amplitude) which could then feedback into the circuit. This is neither shown nor discussed in the manuscript. It would be easy to compare the “expected” temporal profile of two reporters in the different oscillation scenarios (without having these reporters interfere with the circuit in simulations). This would at least add some credibility to this claim. Additionally, to support such an unexpected claim, verifying it experimentally (i.e. seeing that addition of decoy AraC binding sites does not rescue the oscillations) would be important. Finally, there should be at least some discussion to explain this effect if there is no theoretical support.

=> We appreciate all of these comments and the suggestion to polish our manuscript. For clarity, we define “expression rate” and “expression level” as separate concepts. The latter is a time integral of the expression rate and the degradation/dilution rate.

Hence, when our in vivo system has sufficient and constant concentrations of AraC and arabinose and lacks LacI, our reporters A and B have similar maximum expression levels (Figure 5), and thus, similar expression rates. In our simulations, we set the same maximum expression rate for both reporters A and B.

Regarding the expression rate and e expression level shown in Supplementary figures 5 and 13 and the sentence “comparison of the two the expected temporal profile of two reporters in the different oscillation scenarios (without having these reporters interfere with the regulatory circuit in simulations),” note that different profiles are observed due to differences in the dependencies of regulatory proteins.

We hope that this additional supporting figure and the descriptions in the Discussion section (line 277) provide satisfactory support for our claim, and that the reviewer agrees that additional experiments are not required.

Page 15, line 412 (Results and Discussion)

Despite identical maximum transcription rates, each of the present downstream promoters showed different temporal expression profiles of the reporter protein during the same oscillation period (Figure S5); this oscillation period was similar to that derived from simulations that assume no interference of the reporters with the regulatory circuit. This difference in activation timing reflects relative dependencies of the regulatory proteins. In other cases with retroactivity and/or resource sharing, differences in temporal profiles of the reporter protein from regulatory circuits produces further differences in all features of oscillation.

Supplementary Page 30 (Figure S5)

Other major issues:

-Language and clarity

Although the English is technically correct, it is extremely hard to understand. It should be almost completely rewritten. It is unclear throughout the paper what is being shown, what system is being studied, and even the definitions.

=> We apologize for our unclear English writing. We have revised the manuscript to improve on this. For clarity, we define Smolen oscillator in the introduction as follows:

Page7, Line 122 (Introduction)

We modeled and simulated alternative downstream genes in the Smolen oscillator, which forms positive and negative feedback loops (Figure 1A) [11,29,30]. These computations indicate that competition from protein-binding sites to their regulatory proteins and from target proteins to their tag-specific proteases can alter the dynamics of cellular systems. These two process are respectively consistent with the previously proposed concepts of retroactivity [13-15] and competition between target proteins and their tag-specific proteases [26,31].

-Abstract

The abstract does not state the results, any conclusion, what kind of synthetic circuit is being investigated, does not define retroactivity and resource competition, etc.

=> We thank the reviewer for these helpful comments. We have revised the abstract as follows:

Page 2, line 18 (Abstract)

In studies of synthetic biology, analyses of reporter gene activities ensure control of the cell with synthetic genetic circuits, as achieved using a combination of in silico and in vivo experiments. However, unexpected effects of downstream reporter genes on upstream regulatory genes may interfere with in vivo observations. This phenomenon is termed as retroactivity.

Page 2, line 25 (Abstract)

Furthermore, our models demonstrated that differences in resource competition for a protease that degrades regulatory and reporter proteins affect changes in upstream regulatory circuits, even when the circuits share the same number of regulatory-protein binding sites.

-Organisation

The organisation of the paper could be improved. It appears to have been arranged chronologically, but it is confusing for the reader. The way it is written now, it suggests something and then say it is wrong later and therefore section 3.1 is not necessary. I would suggest starting with the experiments (3.2) then going into the detailed simulations teasing apart the contribution from retroactivity and resource competition. Section 3.4 does not bring anything to the manuscript and should be removed.

=> We apologise for our clumsy descriptions in section 3.1 and Figure 4, and fear that these have caused confusion. To clarify the necessity of this section and its figure, we have revised the text in lines 260–264 as follows:

Page 15, line 257 (Results and Discussion)

For some sets of parameters, the variables for numbers of molecules per cell fell below 1. Thus, the stochastic dynamics of molecules possibly regulate oscillations in our living cells. In Smolen’s oscillators, however, stochastic Gillespie and deterministic simulations showed similar periods for a set of parameters [11,44].

=> We contend that Section 3.4 is important for future genetic circuit designs. To demonstrate its significance, we now include comparison of genetic and electric circuits in the introduction section (Line 135).

Page 8, line 139 (Introduction)

In further designs of genetic circuits, we included resource competition and compared it in electric circuits using established design strategies.

-Figures

Figure 1 is not useful and almost impossible to understand. It would be better to merge it with figure 3 and provide a schematic of the different conditions tested. Figure 2 also does not bring anything and should be removed. In figure 3, it is unclear that the proteins are targeted for degradation from the schematic. The design of the figures could also be improved. The subheading (e.g. A,B,C) are too large and the axis labels too small to read. Figure 8 should be removed. Exporting the figures as vector graphics would also help.

=> We thank the reviewer for these suggestions and have revised Figure 1 accordingly, firstly to introduce the previous figure 3, which is a schematic of test conditions. The revised figure also includes comparisons of binding sites to show retroactivity in the original Figures 1 A–C. To introduce resource competition, we now present the original Figure 1B as Figure 2, and moved the original Figure 2 to the supplementary information. The subheading in figure 1 has been reduced in size and the axis labels have been enlarged. For better understanding of section 3.4, which has been revised for clarity in the revised manuscript, we believe that figure 8 is important.

Page 5, line 72 (Introduction)

Page 6, Line 93 (Introduction)

There also might be potential issue with the copyrights of some figures. Figure 3 and Figure 8 are very similar to figures published in a previous article by the same first and last authors (Moriya et al. BMC Systems Biology 2014).

=> We appreciate the reviewer’s comments. In the BMC Systems Biology website, it is stated that the Creative Commons attribution license permits unrestricted use, distribution, and reproduction in any medium, provided the original work is properly cited. With adequate references in this manuscript, we believe we have avoided any potential copyright issues.

-Experiment and data analysis

The measure of “ratio of bottom counts” is unclear and not the best measure of the quality of the oscillations. I think what the author means is that it is the fraction of the cells showing at least x periods (bottom counts = number of throughs in cell traces). It is not a measure of the quality of the oscillations as it depends on the length of the period relative to the short time window in which they are able to follow the cells. Other measurements such as the coefficient of variation in the period (CV, ratio between standard deviation of the period to the mean period) would be more appropriate). The limitations of following cells on agar pad should be discussed (e.g. the growth conditions are changing during the time course). While the experimental methodology is appropriate for this manuscript, a brief discussion of the limitations would be welcomed.

=> We thank the reviewer for these comments. We have revised the description of our experimental limitations in the Results and Discussion section as follows:

Page 12, line 358 (Results and Discussion)

In the same microcolony on an agar plate, the physiological conditions of cells, even in a rim region of the microcolony, varied with time probably because of cellcell communications or nutrient deficiency. Accordingly, in our experiments with long incubation periods, we could not detect oscillations in most microcolonies, even when these microcolonies included cells with oscillations at earlier time points (Supplementary Movie 1). Thus, our microscopy experiments were limited to 3 h. Due to this limitation, we were not able make other measurements, such as those required to estimate coefficients of variation in the period.

In conclusion, while the results are not ground-breaking, they could be of interest to the synthetic biology community. If it is true that a slight variation in the dynamic profile of a reporter can have drastic effect on gene circuits, then it would be quite interesting. However, to reconsider this manuscript, the authors should provide more support to their claim (either experimentally or theoretically, preferably both), rewrite the manuscript and clarify the figures.

=> We appreciate the reviewer for providing valuable comments and hope that our revisions support our claims clearly.

Reviewer 2 Report

In their manuscripts the authors analyzed one of the dark corners of synthetic biology: the upstream effect of a reporter on the working of a circuit. Despite the limited applicability of the results to other circuits, the take-home message on the importance of including retroactivity and resource competition in the modelling is relevant enough for me to recommend its publication.

Here are some comments:

- Overall the readability of the manuscript could be improved. In particular I would make more obvious that the aim of the paper is to study the oscillator of Stricker et al., which proved to be a robust one, and question the possible upstream roles of the reporter. This lack of clarity makes difficult to understand at the beginning what kind of circuit are the authors studying. The general message on more broad circuits that is addressed in the conclusions would still be valid

- It would be very useful to have Figure 3 earlier in the text. Probably as part of Figure 1. That would make more clear the topology of the network studied from the first moment. As it is now, Figure 1 is very difficult to follow. Adding names on top of the circuits of Figure 1, or adding labels such as "decoy", or "protease" would help as well.

- Figure 2 does not have any use at all. Along the manuscript there is no reference to the promoter states, making the figure not readable without understanding the mathematical model, or without having any kind of label in the Figure. I would either move this figure to the supplementary part with the model. Alternatively, it would also be interesting to make an effort in making FIgure 2 more skematic including the different possible reactions in a clear way.

- I might have missed something on the details of the promoter, but why the affinity of Arabinose binding site is dependent on IPTG? And why those affinities depend on the same parameters as the LacI ones (kr1,b1). Additionally in table A1, there is a constant b2 that I did not found in the model description, is this a typo?

- When thinking on limiting rate reactions I find useful to think on the minimum numbers of molecules of each kind. This information is not clear in the manuscript. For instance in Figure 4 is hard to see which is the minimum number of molecules, how they compare to the number of binding sites, and how many molecules are bound at different timepoints. Would having a logscale help?

- Additionally, if we are in a limit where having very few copies might be important, it would be good to know this lower limit to make sure that the deterministic description is correct (this also includes numbers of mRNA).

- Finally, one of the main conclusions of the paper seems to be the role of timing. In particular, oscillations in the system depend on a fine tuning of the timing of degradation/production of the different proteins and mRNAs. If protein degradation is too slow, oscillations are destroyed. This is in line with recent theoretical studies that show that more careful functions for degradation are necessary in the modelling (J. Tyler [arXiv:1808.00595v1] et al.), or showing that oscillations are destroyed when the protein degradation is too slow (K.M. Page et al. Royal J.Soc. Interface 2018). In this aspect, the "competition for resources" would not be limiting in terms of "energy" but in terms of how different are the dynamics of the protein compared to those of faster processes such as the mRNA.

Author Response

Note:

Blue-coloured text = the reviewers' comments

Black-coloured text = responses to the comments (starting with '=>') or unrevised text Underlined black-coloured text = text surrounding revisions

Green-coloured text = revised text

Underlined green-coloured text = revised text with correlation

 ------------------------------------------------------------------------------------

Responses to Reviewer.

In their manuscripts the authors analyzed one of the dark corners of synthetic biology: the upstream effect of a reporter on the working of a circuit. Despite the limited applicability of the results to other circuits, the take-home message on the importance of including retroactivity and resource competition in the modelling is relevant enough for me to recommend its publication.

=> We appreciate the reviewer’s understanding of our study and are thankful for these comments.

Here are some comments:

- Overall the readability of the manuscript could be improved. In particular I would make more obvious that the aim of the paper is to study the oscillator of Stricker et al., which proved to be a robust one, and question the possible upstream roles of the reporter. This lack of clarity makes difficult to understand at the beginning what kind of circuit are the authors studying. The general message on more broad circuits that is addressed in the conclusions would still be valid.

=> We agree that the Smolen oscillations reported by Stricker et al. are a central focus of this study. To make this focus more clear, we have added text to this effect in the introduction, and we combined the former Figures 1 and 3 and added protein names and network topologies.

Page 2, line 22 (Abstract)

Using in silico and in vivo experiments, we found that a different copy number of regulatory protein-binding sites in a downstream gene altered the upstream dynamics, suggesting retroactivity of reporters in this synthetic genetic oscillator.

Page 4, line 57 (Introduction)

The genetic circuits shown in Figure 1 A-D share upstream genes for which topological characteristics are favourable for oscillations. These circuits, however, differ in their reporter genes, which also have binding sites for regulatory proteins (Fig 1 E-H).

Page 5, line 72 (Introduction)

Page 6, line 93 (Introduction)

Page 7, line 122 (Introduction)

We modeled and simulated alternative downstream genes in the Smolen oscillator, which forms positive and negative feedback loops (Figure 1A) [11,29,30]. These computations indicate that competition from protein-binding sites to their regulatory proteins and from target proteins to their tag-specific proteases can alter the dynamics of cellular systems. These two process are respectively consistent with the previously proposed concepts of retroactivity [13-15] and competition between target proteins and their tag-specific proteases [26,31].

- It would be very useful to have Figure 3 earlier in the text. Probably as part of Figure 1. That would make more clear the topology of the network studied from the first moment. As it is now, Figure 1 is very difficult to follow. Adding names on top of the circuits of Figure 1, or adding labels such as "decoy", or "protease" would help as well.

=> We thank the reviewer for this comment. To improve clarity, we combined Figures 1 and 3 and revised the manuscript and figure according to this suggestion.

- Figure 2 does not have any use at all. Along the manuscript there is no reference to the promoter states, making the figure not readable without understanding the mathematical model, or without having any kind of label in the Figure. I would either move this figure to the supplementary part with the model. Alternatively, it would also be interesting to make an effort in making FIgure 2 more skematic including the different possible reactions in a clear way.

=> Thank you for your suggestion. We moved Figure 2 to the supplementary material.

- I might have missed something on the details of the promoter, but why the affinity of Arabinose binding site is dependent on IPTG?

=> We agree with the question raised by the reviewer after having read the results of IPTG binding rates at the AraC protein reported in a previous study (Stricker 2008). Accordingly, we found that IPTG is a reported inhibitor of the PBAD expression system (Lee, Appl Environ Microbiol. 2007), and, thus, in the present study, we described the binding rates at the AraC protein to accommodate not only arabinose but also IPTG in our model.

Page 9, line 153 (Materials and Methods)

Binding rates (ka) at the AraC protein binding site were determined in a previous study, and these indicated that IPTG weakly inhibits arabinose-binding to AraC [11,36].

And why those affinities depend on the same parameters as the LacI ones (kr1,b1). Additionally in table A1, there is a constant b2 that I did not found in the model description, is this a typo?

=> We sincerely appreciate this comment. Two typographical errors were present in formula (2), which represents binding constants of AraC to the promoters shown by Stricker 2008. To address this, we revised the terms kr1 and b1 to kr2 and b2. These parameters were present in our simulation code with the parameters quoted from Stricker 2008. Moreover, b2 is shown in Appendix Table A1 of the original submission. Although we included kr2 in our simulations, we failed to describe this parameter in Appendix Table A1 and have now added it to the revised table. In the report by Stricker 2008, IPTG binding to AraC was weaker than that to LacI and kr2 was correspondingly 50 times higher than kr1.

Supplementary Page 5, line 81

Page 36, line 10 (Appendix Table A1)

kr2

1.8

correction constant of

arabinose

mM

[11]

- When thinking on limiting rate reactions I find useful to think on the minimum numbers of molecules of each kind. This information is not clear in the manuscript. For instance in Figure 4 is hard to see which is the minimum number of molecules, how they compare to the number of binding sites, and how many molecules are bound at different timepoints. Would having a logscale help?

=> We appreciate this suggestion. In our revised figure 4, log-scaled graphs show that some variables fall below 1. Thus, we conclude that the stochastic dynamics of molecules possibly regulate oscillations in living cells. In Smolen’s oscillator, however, both stochastic Gillespie and deterministic simulations showed similar periods for a set of parameters [11,44].

Page 15, line 257 (Results and Discussion)

For some sets of parameters, the variables for numbers of molecules per cell fell below 1. Thus, the stochastic dynamics of molecules possibly regulate oscillations in our living cells. In Smolen’s oscillators, however, stochastic Gillespie and deterministic simulations showed similar periods for a set of parameters [11,44].

Page 17, line 274 (Results and Discussion)

- Finally, one of the main conclusions of the paper seems to be the role of timing. In particular, oscillations in the system depend on a fine tuning of the timing of degradation/production of the different proteins and mRNAs. If protein degradation is too slow, oscillations are destroyed. This is in line with recent theoretical studies that show that more careful functions for degradation are necessary in the modelling (J. Tyler [arXiv:1808.00595v1] et al.), or showing that oscillations are destroyed when the protein degradation is too slow (K.M. Page et al. Royal J.Soc. Interface 2018). In this aspect, the "competition for resources" would not be limiting in terms of "energy" but in terms of how different are the dynamics of the protein compared to those of faster processes such as the mRNA.

=> To address this comment we have revised the manuscript to include descriptions of degradation timing. We don’t think that energy consumption and the timing of degradation are opposing concepts. Rather, in combination, these concepts are important contributors to considerations of oscillator dynamics.

Page 31, line 498 (Results and Discussion)

In very recent studies, oscillator dynamics were shown to depend on the fine-tuning of product degradation pathways [48,49]. Controlled timing of degradation may require more energy than maintenance of constant expression levels of degradation enzymes. Questions arising from this hypothesis will be important subjects of future studies.

Round 2

Reviewer 1 Report

See attached file

Author Response

Note:

Blue-coloured text = the reviewers' comments

Black-coloured text = responses to the comments (starting with '=>') or unrevised text Underlined black-coloured text = text surrounding revisions

Green-coloured text = revised text

Underlined green-coloured text = revised text with correlation

 ------------------------------------------------------------------------------------

Responses to Reviewer.

The authors made minor changes to the manuscript to help address some of my concerns. I think the manuscript could be published with minor modifications, although in the present form it would not be as useful to the community as it could be. I will leave it to the editor to decide what are the minimum requirements in terms of presentation for publication. The main barrier is the text of the manuscript. It is very hard to read and very long. There are many sentences that repeat exactly the same information right next to each other. As I commented before, the manuscript would benefit from being rewritten entirely in a more succinct manner. I think it could be half the size while being clearer and easier to understand.

=> We appreciate the reviewer for understanding our study and giving us a variety of useful comments and suggestions for this manuscript.

Regarding my previous concerns:

Different promoters and effects of protease competition Figure S5 does help to see that despite the lac and lac/ara promoter having similar maximum expression, they can have very different expression profiles (as seen in the almost 5-fold difference in the level of the proteins). This should be clarified in the text, and it should be clear that it is the level of the proteins that matter for competition of proteases, and that this is vastly different between the promoters. It would also be important to have a figure showing the repression/activation curve of the two promoters to show that their differences (for example as 2D heatmap, [lac] vs [ara]). It would help clarify the very sharp non-linearity of the dark blue line in figure S5. This and/or S5 should be in the main text (instead for example figure 7), as this is the main conclusion from the paper (i.e. different activation/repression curves for the reporter can result in different levels of the reporter even with similar maximum transcription, which in turn result in different effects on protease competition). Also, the text should mention that the maximum rates are similar, not identical (there is a ~30% difference).

=> We thank the reviewer for these suggestions. We moved Figure S5 to main figure. Moreover, we added 2D heatmap of the two promoters between AraC and LacI.

 We apologise for painful expression rate curve. For your suggestion, we correct this expression rate curve to make it easy to read. For the readability, we thank J. Noack for English writing.

Page26, line 408

In other words, we added simulations without competition from protein-binding sites to their regulatory proteins and/or that from target proteins to their tag-specific protease. Without both of these competitions, all three circuits showed the same oscillation area and regulatory-protein amplitudes (Figures 6J–L and S10J-L), although the lac/ara-reporter circuit showed a lower GFP amplitude in expression level than the others (Figure S12J-L).

Page27, line 428

The large effect produced by protease-sharing can be explained in detail when we focus on a set of IPTG-arabinose concentrations. Although all circuits showed the same oscillation period, the much higher temporal GFP expression level profile of the lac promoter than that of the lac/ara promoter accounts for the large effect produced by protease-sharing (Figure 7A). This difference in expression level was derived from different temporal expression rate profiles of the two promoters (Figure 7B) whose expression rates depend on the concentration of regulatory proteins (Figure 7C, D). Despite the two promoters in the simulation having the same maximum transcription rate parameter (ka and kr), AraC concentration during the oscillation period is not enough to achieve the maximum expression rate from the lac/ara promoter.

Page 35, line 568

We thank J. Noack for English writing.

Page 27, line 439

Figure 7Oscillation timing of the three synthetic circuits.

Deterministic simulations performed at arabinose 1.0% and IPTG 0.01 mM without downstream competitions from protein-binding sites to their regulatory proteins and from target proteins to their tag-specific proteases. The amount of active promoter of the lac/ara-reporter circuit model (black), the lac-reporter circuit model and the lac-reporter + AraC decoy circuit model (blue).

(A) GFP Expression level of the three synthetic circuits.

(B) Expression speed performed by the amount of GFP active promoter.

(C) The lac/ara promoter state with black-lined trajectory at arabinose 1.0% and IPTG 0.01 mM.

(D) The lac promoter state with black-lined trajectory at arabinose 1.0% and IPTG 0.01 mM.

- Abstract

The abstract has been slightly improved, but it still does not mention the actual synthetic circuit being studied. The conclusion should also be stated in simpler terms. E.g.

By separating the two sources of retroactivity (titration of the component and competition for degradation), we showed that, in the dual-feedback oscillator, the level of the fluorescent protein reporter competing for degradation with the circuits’ components is important for the stability of the oscillations.”

=> We thank the reviewer for these suggestions. We inserted your suggestion in our manuscript.

Page 2, line 25

Furthermore, by separating the two sources of retroactivity (titration of the component and competition for degradation), we showed that, in the dual-feedback oscillator, the level of the fluorescent protein reporter competing for degradation with the circuits’ components is important for the stability of the oscillations.

Minor points:

-Typo at line 327 – “mode”

=> We apologise for this typographical error in line 247.

Page 14, line247

In contrast, we recently demonstrated the effects of retroactivity and protease sharing by the addition of the reporter protein using mathematical model [12].

-In section 3.1, it would help clarify the paper if it was clearly specified that both titration of components and different reporter dynamics/ levels of protease competition are present

=> We thank your suggestion. We inserted in two competition description in section 3.1.

Page15, line273

This is an example of in silico retroactivity [13-15] and/or resource competition [26,31], in other words, titration of the component and/or competition for degradation, respectively.
